# Salinity-Triggered Responses in Plant Apical Meristems for Developmental Plasticity

**DOI:** 10.3390/ijms24076647

**Published:** 2023-04-02

**Authors:** Soeun Yang, Horim Lee

**Affiliations:** Department of Biotechnology, Duksung Women’s University, Seoul 03169, Republic of Korea; dosl68997719@duksung.ac.kr

**Keywords:** abiotic stress, developmental plasticity, root apical meristem, salt tolerance, shoot apical meristem

## Abstract

Salt stress severely affects plant growth and development. The plant growth and development of a sessile organism are continuously regulated and reformed in response to surrounding environmental stress stimuli, including salinity. In plants, postembryonic development is derived mainly from primary apical meristems of shoots and roots. Therefore, to understand plant tolerance and adaptation under salt stress conditions, it is essential to determine the stress response mechanisms related to growth and development based on the primary apical meristems. This paper reports that the biological roles of microRNAs, redox status, reactive oxygen species (ROS), nitric oxide (NO), and phytohormones, such as auxin and cytokinin, are important for salt tolerance, and are associated with growth and development in apical meristems. Moreover, the mutual relationship between the salt stress response and signaling associated with stem cell homeostasis in meristems is also considered.

## 1. Introduction

Salinity is a severe environmental problem that adversely affects the growth, development, and productivity of important crops. Recent environmental issues, such as climate change, have increased evaporation and salt contamination through high temperatures and sea-level rises, respectively, exacerbating salt stress effects on plant growth and development [1]. Soil salinization has expanded continuously to approximately 20% of the World’s agricultural lands. Indeed, salt stress causes economic problems in agriculture [2]. Therefore, it is crucial to understand how plants tolerate and adapt to salt stress for growth and production.

Since salt stress is usually caused by the increased concentration of ionized sodium (Na^+^) and chloride (Cl^−^) ions, salt stress affects plant responses along with osmotic and ionic stresses [3]. Osmotic stress is induced rapidly by the low water potential caused by high salt concentrations in the soil, showing a significant decrease in the rate of shoot growth [4]. Ionic stress induces a slower response due to the internal accumulation of salt ions, which are absorbed from the roots and transported to the shoots [4]. These two responsive phases under salt stress conditions also trigger intracellular stress signals, including reactive oxygen species (ROS), which cause oxidative damage to macromolecules and give rise to defects in plant growth and development [3,5].

Salt stress increases the cytosolic Ca^2+^ concentration [6,7]. A recent finding showed that the process related to salt stress-increased concentrations of cytosolic Ca^2+^ is performed by glycosyl inositol phosphorylceramide (GIPC) sphingolipids in the plasma membrane [8]. In this study, the cytosolic increase in Ca^2+^ triggered by salt stress was blocked in the *monocation-induced [Ca^2+^]_i_ increases 1* (*moca1*) mutant, which contains the *MOCA1* loss-of-function mutation that encodes a glucuronosyltransferase involved in the biosynthesis of plant-specific GIPC sphingolipids [8]. These results show that GIPC is a salt stress sensor that binds to Na^+^ ions and opens Ca^2+^ influx channels.

Salt stress-induced cytosolic Ca^2+^ levels have been known to activate Salt Overly Sensitive 3/SOS3-like Calcium-Binding Protein 8 (SOS3/SCABP8) encoding of an EF-hand calcium-binding protein [9]. Activated SOS3/SCABP8 interacts with the serine/threonine protein kinase SOS2 for sequential activation [10,11]. Activated SOS2, then, regulates the downstream SOS1 encoding of a Na^+^/H^+^ antiporter required to pump cytotoxic Na^+^ ions out of the cells [12]. These sequential activations of the SOS proteins in the SOS pathway were insensitive in the *moca1* mutant [8], suggesting that the GIPC as a salt sensor connects the gap between salt stress and the SOS pathway.

Eventually, the SOS pathway can be seen as a process of developmental plasticity related to salt-mediated growth, via the detoxification of ionic stress, induced under salt stress conditions. As physiological and morphological responses, developmental plasticity is one of the adaptation or tolerance strategies through the plastic properties, in which a single genotype shows a wide range of responses against unfavorable stresses on plant growth and development [13]. Through in vitro shoot apex culture using 15 different genotypes in orange-fleshed sweet potato (*Ipomoea batatas* L.), growth retardation of shoots and roots was found to be correlated with increased NaCl stress conditions [14]. Among them, six sweet potato genotypes with relatively tolerant growth showed significantly increased levels of antioxidant enzymes, including superoxide dismutase (SOD), catalase (CAT), and guaiacol peroxidase (GPX), under NaCl stress conditions [14]. The effect of antioxidant enzymes to scavenge ROS is also known to be important for salt-tolerant growth in other plant species, such as peas and wheat [15,16]. These results suggest that plastic plant growth under salinity is probably correlated with the regulation of endogenous ROS levels through antioxidant enzymes. Since plants develop postembryonically, primary apical meristems replenishing new daughter cells to form new lateral organs seem to be an essential place for the plastic adaptation of growth and development during salt stress conditions. In this review, we will discuss the recent findings on salt-mediated acclimation mechanisms focusing on primary apical meristems for developmental plasticity.

## 2. Developmental Plasticity via the Primary Apical Meristems under Salt Stress Conditions

### 2.1. Salt-Triggered Plastic Growth and Development in Root Meristems

Roots are the primary organs facing a soil environment containing unfavorable abiotic stresses, including high salt concentrations. Therefore, the root system architecture (RSA) shows plastic traits for proper root growth and development in response to exogenous nutrients and abiotic stress factors [17]. For example, tomato seedlings showed a significant decrease in the lower apical zone in the root tip and an increase in the upper basal zone connected with hypocotyls in the shortened main roots, under salt stress conditions [18]. Owing to the natural property of postembryonic development, root plasticity mainly relies on the activity changes in the root apical meristem (RAM) to regulate stem cell proliferation and differentiation.

Previously, root developmental genes, such as *SHORT ROOT* (*SHR*) and *SCARECROW* (*SCR*), were reported to activate microRNA (miRNA) 165/166 to degrade its target mRNAs, encoding the class III homeodomain-leucine zipper transcription factors (HD-ZIP III TFs), and determining the cell fate for xylem patterning in root meristems [19]. In addition, the regulatory feedback loop consisting of *PHABULOSA* (*PHB*), one of five HD-ZIP III TF family members, and cytokinin (CK) determine the balance between stem cell proliferation and differentiation to regulate root growth [20]. Interestingly, a recent finding showed that the morphological adaption of *Arabidopsis* RSA under salt stress conditions is modulated by miRNA165/166 in the roots [21]. In this study, the salt-mediated reduction of miRNA165/166 increased *PHB* expression and the *ISOPENTENYL TRANSFERASE 7* (*IPT7*) expression-mediated CK levels. These results suggest that the regulation of miRNA165/166, via *PHB* and CK, is required for the plastic adaptation of root growth and development in response to salt stress conditions because the *phb-1d* mutant harboring a mutation in the miRNA165/166 target site of *PHB* gene, while the *ipt7* mutant related to CK biosynthesis showed the resistant phenotypes to the salt stress-mediated reduction of root growth [21]. Since the regulatory circuit of miR165/166 and HD-ZIP III TFs, including PHB, is well known to be involved in regulating shoot and lateral organ development [22], it will be interesting to examine whether the process of plastic adaptation, mediated by miRNA165/166 under salt stress conditions, is also applied to shoot growth adaptation.

The function of redox status (potential) is important for plant growth and stress response via the change in ROS level and distribution [23,24]. The redox-signaling hub consisting of oxidant ROS and antioxidants can interact with phytohormones, including auxin, which is involved in the regulation of cell division/elongation/differentiation, and the increase of tolerance to biotic/abiotic stress, for plastic plant growth adaptation against environmental stress [25]. For example, it has been reported that salt stress-mediated root growth inhibition correlates with the redox status change in *Arabidopsis* root meristems [26]. In this study, the root stem cell niche, including the quiescent center (QC) and proximal meristem (PM: also known as division zone), exhibited the most reduced redox status, which rose to a more oxidized status in the transition zone (TZ), ceasing cell division in the roots, under optimal conditions. In contrast, the overall redox status of QC and PM was changed to a more oxidized status under salt stress conditions [26]. The function of the redox potential has long been proposed as being linked to cell division activity [27]. In addition, the transcriptional regulation of ROS has been reported to control the balance between cell proliferation and differentiation through the basic helix-loop-helix TF UPBEAT1 (UPB1), which directly regulates the expression of a set of peroxidases that modulate the ROS distribution in the roots [28]. Moreover, the region of the most reduced redox status of QC also overlapped with that of the auxin maxima, under optimal conditions [26,29], while the auxin signal represented using *DR5rev:GFP* transgenic line was significantly reduced by the treatment of salt stress conditions [29]. Since auxin has been reported to play an important role in developmental plasticity under abiotic stress conditions, including salinity, and promote ROS production via the activation of NADPH oxidase [25,30], these studies suggest a correlation between the redox status and auxin regulation for adaptive root growth under salt-stress conditions.

Previous studies supported the idea that crosstalk between salt stress-mediated ROS and auxin signaling is required for growth acclimation to salinity [31,32]. Iglesias and colleagues showed that salt stress conditions suppress auxin signaling through the post-transcriptional inhibition of auxin receptors, such as *TRANSPORT INHIBITOR RESISTANT1* (*TIR1*) and *AUXIN SIGNALING F-BOX* (*AFB*), by salt stress-induced *miR393* expression [32]. As expected, under salt stress conditions the reduced *TIR1* expression was alleviated in the *ago1-27* and *mir393ab* mutants, where the miRNA biogenesis was disrupted [32]. In addition, the salt-mediated inhibition of root growth was less reduced in the *mir393ab* and *tir1 afb2* mutants than in the wildtype (WT) [31,32]. These findings suggest that the complex network, including salt stress-triggered miRNAs, redox status, and auxin changes in root meristems, is important for plastic root growth and development under salt stress conditions.

Nitric oxide (NO) plays a versatile role in plant growth and development, including the roots [33]. For example, the NO signal, leading to decreased cell division and promoted cell differentiation in root meristems, is accumulated by abiotic stimuli, such as salt stress [34]. In addition, applying a NO biosynthetic inhibitor alleviates the salt-mediated growth inhibition of root meristems [35]. Moreover, the highly accumulated NO levels, which were observed in *chlorophyll a/b binding protein underexpressed1*/*NO overproducer1* (*cue1*/*nox1*) mutants, are related to the reduced acropetal auxin transport and maxima via a decrease in the protein levels of the efflux carrier PIN-FORMED1 (PIN1), which leads to the disorganization of QC and columella stem cell (CSC) [34,35]. In this process, the low activity of auxin signaling caused by NO accumulation was enhanced by the stabilization of the auxin signaling repressor IAA17 [35]. The polar auxin transport by influx/efflux carrier proteins, localized at a specific membrane domain, is important to establish auxin maxima in root meristems [36]. A recent study reported that PIN proteins are regulated by endocytic recycling to restrict carriers at the specific membrane domains without lateral diffusion through the Rho-of-plant 2 (ROP2) GTPase [37]. This study also showed that the ROP2 GTPase is necessary for NO-mediated root growth inhibition. When NO donors, such as GSNO or SNAP, were applied exogenously, the *rop2-1* mutant showed less sensitive root inhibition of the primary root length than the WT [37]. Thus, these results suggest that PIN1 modulation of the auxin accumulation, by regulating ROP2 GTPase-mediated endocytic trafficking in root meristems, is essential for NO-mediated root growth inhibition. Previously, NO has been known to affect the regulation of RSA via post-translational modification, such as the *S*-nitrosylation of target proteins, including a plastidial glyceraldehyde-3-phosphate dehydrogenase (GAPDH), via a reduction of enzyme activity [38,39]. Interestingly, the reduced root growth and meristem size caused by the higher levels of NO shown in the *cue1*/*nox1* mutants were partially rescued by the exogenous application of glycine and serine, which are downstream products of the plastidial GAPDH [38], suggesting that the reduced root growth is probably due to the decreased enzyme activity of plastidial GAPDH, as modified by NO.

Similarly, as mentioned earlier, ROS signaling associated with auxin is also important for controlling RSA [25,39]. NO and ROS act as signaling molecules to up/down-regulate the expression of genes involved in auxin biosynthesis and signaling to control RSA. In addition, ROS, such as hydrogen peroxide H_2_O_2_, mediates auxin homeostasis by regulating polar auxin transport [40]. A recent finding has shown that altered ROS levels by a ROS source (H_2_O_2_) or ROS production inhibitor (diphenylene iodonium) lead to changes in auxin distribution via PIN proteins, which in turn regulate seedling root growth [41]. Consistently, mutants of *root hair defective2* (*rhd2*) and *superoxide dismutase* (*sod*) that mimic altered ROS balance have been found to exhibit changes in auxin distribution [41]. These results suggest that ROS balance-mediated auxin distribution and the response in root meristems are important for root growth and development. Interestingly, it has been reported that NO and ROS interact to regulate the biosynthesis and protein activity of antioxidant enzymes for abiotic stress responses [42]. For example, the cytotoxic ROS levels induced by abiotic cadmium stress inhibiting root growth were detoxified by elevated NO levels regulating the activity of antioxidant enzymes, such as SOD, ascorbate peroxidase (APX), and CAT in *Brassica juncea* [43]. Similar to the role of NO, methyl jasmonate (MeJA) also showed the alleviation of salt-mediated growth inhibition associated with the mitotic index of RAM cells and the biomass of shoot/root in bitter almonds [44]. In addition, the exogenous application of MeJA in optimal concentrations was revealed to mitigate the growth defects by increasing the rate of photosynthesis and the activity of antioxidant enzymes, including APX, SOD, and GPX [44]. Although the cooperation of each NO or ROS with phytohormones, such as auxin and MeJA, has been revealed to regulate RSA based on previous studies, the direct evidence of crosstalk among NO, ROS, and phytohormones for adaptive plastic root growth under salt stress conditions remains elusive.

### 2.2. Plastic Growth and Development in Shoot Meristems Triggered by Abiotic Stress Including Salt

The shoot apical meristem (SAM) contains undifferentiated stem cells, which are divided continuously into self-renewal stem cells and specified daughter cells, which replenish the developmental pattern formation of the aboveground parts of adult plants after embryogenesis [45]. Since plants are not mobile, shoot growth and development, supported by stem cells in the SAM, must be acclimated for phenotypic plasticity against unfavorable external and internal stress conditions [46]. For example, the mechanical stress conditions caused by tissue folding that occurs during developmental patterning were reported to promote the expression of *SHOOT MERISTEMLESS* (*STM*), which encodes a homeodomain TF and functions in the initiation and maintenance of shoot meristems [47]. In animals, the mechanical stress signal is also known to be important for controlling stem cell fate, including self-renewal and differentiation [48]. In addition to the SAM activity, STM is also involved in organ separation in the SAM. Further, more enhanced *STM* expression was found at the curvature of the boundary domains, where the mechanical stress is established, and between the undifferentiated shoot meristem and flanking the differentiated primordia for boundary formation, where it performs the role of interaction between miRNA164 and its target *CUP SHAPED COTYLEDON* (*CUC*) genes [47,49]. Another study also reported that the expression of several tomato *GRAS* genes involved in fundamental processes of plant growth and development was increased significantly by mechanical and abiotic stress conditions [50]. In addition, mechanical stress has been revealed to induce biotic and abiotic stress responses via transcriptomic analysis [51]. Moreover, it was proposed that integrating mechanical stress and plant hormones, including CK and auxin, is important for SAM growth and development [52]. Despite the direct evidence of interactions between mechanical stress and other environmental stimuli, these results suggest a possible role in mechanical stress-mediated plastic development under abiotic stress conditions.

Maintaining the stem cell population in the SAM is a crucial process for balanced plant growth and development, because the pattern formation of aboveground tissues and organs, regardless of optimal and stress conditions, is supported by the SAM. The stem cell homeostasis in the SAM is controlled mainly through the negative feedback mechanism between CLAVATA3 (CLV3) and WUSCHEL (WUS) [45]. *CLV3* encoding a 12- or 13-amino acid peptide is expressed and secreted from the shoot stem cells [45]. The peptide signal diffusing via the apoplastic space is recognized mainly through the leucine-rich repeat receptor-like kinase (LRR-RLK) CLV1 and BARELY ANY MERISTEM1 (BAM1) receptors on the plasma membrane of neighboring cells [45,53]. Then, receptor-mediated intracellular signaling inhibits the expression of *WUS*, which encodes a homeodomain TF and promotes stem cell proliferation in the SAM [45]. A recent finding showed that CLV3 peptide (CLV3p)-mediated receptor signaling is relayed through mitogen-activated protein kinase (MAPK), MPK3, and MPK6 [53]. In this study, the exogenous treatment of CLV3p caused MAPK activation along the enlarged *clv3-2* SAM. Moreover, the depletion of the shoot stem cell population caused by exogenous CLV3p was significantly suppressed in estradiol-inducible *mpk3–mpk6* double mutants. Therefore, these results suggest that the biological role of MAPK signaling is involved in SAM development.

Intriguingly, it has been revealed that the loss-of-function mutant of *CLV3,* inducing stem cell signaling in the SAM, shows the tolerant phenotype of overall shoot growth under salt stress conditions, compared to the WT [54]. Because the *CLV1* and *BAM1* loss-of-function double mutant acts as CLV3p receptors in the SAM and also displayed an increased survival rate under salt stress conditions [53,54], these results suggest that the CLV3p-CLV1/BAM1-mediated signaling is probably involved in promoting tolerance under abiotic stress conditions, including salinity. The expression of conventional abiotic stress-responsive genes, including *KIN1*, *RD29A*, *RAB18*, and *DREB2A*, was indistinguishable between WT Landsberg *erecta* (L*er*) and *clv3-2*, or WT Columbia-0 (Col-0) and *clv1 bam1* [54], indicating that the salt-tolerant phenotype, shown in stem cell signaling mutants, is caused in a conventional stress response-independent manner. Although the rate of cell division, shown through the 5-ethynyl-2′-deoxyuridine (EdU) staining in the *clv3-2* SAM, was decreased under salt stress conditions, it was similar to the WT L*er* under optimal growth conditions [54]. Therefore, these results indicate that SAM activity represented by the cell division shown in *clv3-2* is presumably vital for the salt-tolerant effect shown in stem cell signaling mutants. It has been reported that moderate heat stress memory (also known as priming) is required for the re-initiation of shoot growth after severe heat stress conditions, via the promoted expression of stem cell regulators, such as *CLV1* and *CLV3*, and the primary carbohydrate metabolism gene *FRUCTOSE-BISPHOSPHATE ALDOLASE 6* (*FBA6*) [55]. In addition, the ectopic expression of *STM* and *WUS* showed increased tolerance to drought stress conditions through the altered activity of stem cell proliferation [56]. Moreover, CLV3/Embryo-Surrounding Region-Related (CLE) peptides, including CLE9, CLE25, and CLE45, were reported to respond, or enhance the tolerance in response to drought, heat, and nutrient deficiency stress conditions [57], and many plant RLKs have been known to be involved in regulating abiotic stress responses [58]. Therefore, these findings suggest that the biological role of stem cell signaling in the SAM is probably necessary for tolerance and adaptation against stress conditions, in addition to SAM maintenance.

In stem cell signaling, CLV3p-mediated CLV1/BAM1 receptor signaling led to the repression of *WUS* expression downstream of *MPK3* and *MPK6*. Eventually, it resulted in the enlarged shoot meristem caused by increased stem cells in the estradiol-inducible *mpk3*–*mpk6* double mutant [53]. Owing to the functional significance of the MAPK cascade, including MPK3 and MPK6, where they serve as an intracellular signaling network involved in biotic/abiotic stress tolerance as well as plant growth [59], these suggest that the function of CLV3p-mediated MAPK pathway regulating stem cell development in the SAM is probably required for growth adaptation and response under stress conditions. Meanwhile, it was reported that tobacco MAPK3 (NtMPK3) interacts with repression of shoot growth (RSG), which encodes a bZIP transcriptional activator to regulate the endogenous levels of gibberellin (GA) for shoot growth, including plant height, internode elongation, and flowering time [60,61]. Point mutations of NtMPK3 phosphorylation sites in the RSG proteins showed the low expression of downstream GA biosynthetic gene, such as tobacco *GA20ox1* (*NtGAox1*), by blocking the nuclear localization of RSG proteins from cytoplasm after the treatment of a bacterial flagellin peptide flg22, which is used as an input signal to activate NtMPK3 [61]. Since NtMPK3 and its *Arabidopsis* ortholog MPK3 are activated by pathogens, these results suggest that MAPK signaling is important for defense-related plant growth [59,62]. In addition, the role of MPK3 and MPK6 was known to be involved in the salt stress response. For example, the phosphorylation of LYST-INTERACTING PROTEIN 5 (LIP5) by MPK3 and MPK6 was reported to be essential for the biogenesis of multivesicular bodies (MVBs) under heat and salt stress conditions [63]. Moreover, salt stress conditions induced the degradation of Arabidopsis response regulator 1 (ARR1), ARR10, and ARR12 proteins via phosphorylation by MPK3 and MPK6 for stress tolerance [64]. For this, MPK3 and MPK6 interacted with ARR1/10/12 and degraded them through the 26S proteasome. Interestingly, ARRs are key regulators in CK signaling, which is involved in plant development, including stem cell homeostasis [65]. In addition, CK negatively controls growth adaptation to high salt stress conditions [66]. Furthermore, it has been revealed that the MAPK kinase 7 (MKK7)–MPK6 module regulates the activity of shoot meristem growth by showing the defect of shoot meristems by the constitutive expression of *MKK7* [67]. Therefore, these findings suggest that the role of MAPK signaling with plant hormones is probably essential for salt-induced stress adaptation in the SAM.

As an osmolyte or a ROS scavenger, proline is known to be highly accumulated in many plant species and protects against salinity-induced osmotic stress [68,69]. Overexpressing two proline biosynthetic genes, *Δ-1-pyrroline-5-carboxylate* synthetase1 and 2 (*P5CS1* and *P5CS2*), in switchgrass (*Panicum virgatum* L.) resulted in increased proline levels, leading to salt-tolerant growth with a reduced level of ROS [70]. In contrast, the loss-of-function *p5cs1* mutant showed hypersensitive phenotypes, such as shortened root length and poor seedling survival to salt stress, with the accumulation of ROS in *Arabidopsis* [71]. In addition to the role in the stress response, Mattioli and colleagues also found that overexpression of *P5CS1* exhibits an early-flowering phenotype compared to the WT [72], suggesting that proline plays an important role in the floral transition. Consistent with this, *p5cs1^−/−^ p5cs2^+/^*^−^ mutants showed a stronger late-flowering phenotype than the WT or the *p5cs1* single mutant [73]. A recent finding has demonstrated that the late-flowering phenotype observed in the *p5cs1^−/−^ p5cs2^+/^*^−^ mutants is due to the significant acceleration of the floral repressor *FLOWERING LOCUS C* (*FLC*) [74]. Given that the control of flowering is a well-known example of developmental plasticity in *Arabidopsis* [13], these findings suggest that proline-mediated regulation serves as a link between salinity and plastic growth in the SAM.

Similar to developmental plasticity in the roots under stress conditions, the redox status caused by the level of ROS is also proposed to be crucial for the developmental process and stress response in shoot meristems [46]. Different types of ROS showed differential distributions in the shoot stem cell niche, where the superoxide anion (O_2_^˙−^) was localized in the central zone (CZ) containing an undifferentiated stem cell population, and H_2_O_2_ accumulated abundantly in the peripheral zone (PZ), occurring differentiation toward specialized cell type to form lateral organs [75]. In addition, these two forms of ROS act antagonistically to control the balance between stem cell proliferation and differentiation for shoot growth through the regulation of ROS-metabolizing enzymes. Consistent with this, H_2_O_2_ as a signaling molecule plays an essential role in stem cell differentiation in animals [76]. More recently, the endogenous stress-related signal (ESS), including stress hormones, was reported to regulate shoot stem cell maintenance, in the stem cell niche via the ETHYLENE-INSENSITIVE 3 (EIN3) TF involved in ethylene signaling under natural growth conditions [77]. EIN3 directly activates the stress regulator *AGAMOUS-LIKE 22* (*AGL22*) in the stem cell niche. AGL22, also known as SHORT VEGETATIVE PROTEIN (SVP), was reported to repress the expression of *CLV1* and *CLV2* [78] and act as a key regulator in gene regulatory networks related to drought tolerance and response [79]. Interestingly, increased *WUS* expression by the treatment of ethylene precursor 1-aminocyclopropane-1-carboxylate (ACC) or polyethylene glycol (PEG) inducing drought stress was blocked in the *svp-41* mutant [77]. Therefore, *AGL22* probably acts as a signaling hub to regulate the plastic postembryonic development in the SAM in response to ESS and exogenous abiotic stress conditions, including salinity.

## 3. Conclusions and Prospects

Salt stress is one of the major environmental stress conditions affecting plant growth, development, and agricultural productivity. Although physiological and molecular responses, including stress sensing, Ca^2+^ flux, SOS signaling, MAPK cascades, and phytohormone signaling, under salt stress conditions, have been studied vigorously [80], a better understanding of plant growth and development will require a change in the current viewpoint toward plastic adaptation on postembryonic growth and development in response to this stress. The morphological development of plants as sessile organisms occur from primary apical meristems regulating the balance of stem cell division/differentiation to maintain stem cells, per se, and differentiate daughter cells for lateral organ formation. Therefore, salt stress-responsive mechanisms related to meristem maintenance are required to understand salt stress-mediated plant tolerance and plastic adaptation. Although recent findings suggest the plausible correlation between signaling mechanisms controlling stem cell homeostasis and environmental/abiotic stress response in primary apical meristems (Figure 1), there is little direct evidence. Therefore, novel molecular genetics and functional genomic approaches focusing on the salt stress response involved in the growth and development of apical meristems will be necessary for understanding how abiotic stress correlates with signaling components, including peptide signals, receptors, and signal transducers, involved in stem cell homeostasis or whether abiotic stress affects developmental plasticity through the same or parallel pathways with stem cell regulation in meristems.

## Figures and Tables

**Figure 1 ijms-24-06647-f001:**
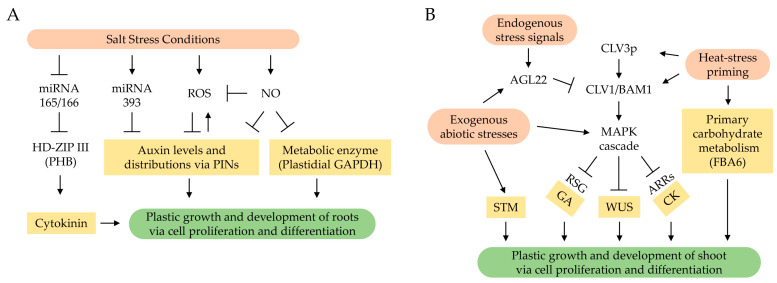
Hypothetical models of signaling networks for stress-mediated plant growth and development. (**A**) Salt stress conditions induce various endogenous signals, such as ROS, NO, and miRNAs, to regulate phytohormones, such as auxin and cytokinin, for controlling adaptive growth and development in root meristems. (**B**) Exogenous and endogenous signals affect various signaling network layers related to the CLV3–WUS pathway regulating stem cell homeostasis in shoot meristems.

## Data Availability

Not applicable.

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
