# Peer review of "Salinity-Triggered Responses in Plant Apical Meristems for Developmental Plasticity"

_ijms, 2023, doi:10.3390/ijms24076647_

Round 1

Reviewer 1 Report

ijms-2305510-peer-review-v1

Yang and Lee's manuscript ‘Salinity-triggered responses in plant apical meristems for developmental plasticity in response to salt stress conditions’ describes an in-depth literature review on apical meristem responses to salinity. The authors have addressed an important topic. The review article is well-compiled; however, I have a few suggestions which could improve the manuscript.

Line 14: Pease specify the phytohormones.

Line 41: ‘eventually causing defects’, please re-write.

The introduction section should contain more literature on developmental plasticity.

In the introduction, please include the salt stress mechanism systematically following biochemical mechanisms, ROS and antioxidant machinery, metabolic action, protein metabolism, and gene regulation. Authors may refer to and cite article 10.1007/s11240-008-9400-2 on sweet potato shoot apex culture under in vitro NaCl-mediated salinity stress conditions.

Line 58-67: Please cite references underlining developmental plasticity.

Section 2.1: The author may highlight the role of root cap cells on salinity tolerance with recent literature. Please reorient the section systematically following ROS accumulation, ROS scavengers and Antioxidative mechanisms, followed by tissue-specific and non-specific gene expression.

Line 87, 106: Many publications focus on salt stress tolerance in other crops apart from Arabidopsis, please refer.

Section 2.2: Please make the sub-headings on ROS accumulation, ROS scavengers and Antioxidative mechanisms, followed by tissue-specific and non-specific gene expression.

Figure: 1A, please revise ‘palstidial GAPDH’ as ‘Plastidial GAPDH’

The authors may include another figure describing Redox signalling-Antioxidative enzyme (Enzymatic and non-enzymatic)-Metabolic action-Gene expression/regulation.

Please illustrate tissue-specific and tissue-non-specific gene regulation mechanisms of apical meristem/root upon salt stress.

The author may describe the osmoprotectants like proline upon salt stress.

The role of elicitors (SA, JA, etc.) on developmental plasticity in response to salt stress may be included.

The prospects of the study/literature review may be elaborated.

References may be arranged as per the journal pattern. Please cross-check the references cited with the list and include more recent relevant references.

Although the overall grammar score is satisfactory, the authors are requested to polish the language, grammar, and punctuation once again.

The manuscript may be accepted with minor corrections.

Good luck with the revision.

Author Response

Dear Editor and Reviewers,

Most of all, we really appreciate the thoughtful reviewers’ comments and suggestions. We have considered them carefully and followed those suggestions to address corresponding concerns and improve our manuscript. We submit two versions of revised manuscripts (marked and clean version) and one revised Figure 1. Detailed our responses to reviewers’ comments and suggestions are mentioned to each comment.

Response to Reviewer 1 Comments

ijms-2305510-peer-review-v1

Yang and Lee's manuscript ‘Salinity-triggered responses in plant apical meristems for developmental plasticity in response to salt stress conditions’ describes an in-depth literature review on apical meristem responses to salinity. The authors have addressed an important topic. The review article is well-compiled; however, I have a few suggestions which could improve the manuscript.

Line 14: Pease specify the phytohormones.

Response: Line14: We appreciate reviewers’ comments! We added specific phytohormones in Abstract as to ‘such as auxin and cytokinin’.

Line 41: ‘eventually causing defects’, please re-write.

Response: We appreciate reviewers’ comments! We re-wrote as “giving rise to defects” instead of ‘eventually causing defects’ at the end of second paragraph in Introduction.

The introduction section should contain more literature on developmental plasticity.

In the introduction, please include the salt stress mechanism systematically following biochemical mechanisms, ROS and antioxidant machinery, metabolic action, protein metabolism, and gene regulation. Authors may refer to and cite article 10.1007/s11240-008-9400-2 on sweet potato shoot apex culture under in vitro NaCl-mediated salinity stress conditions.:

Response: Line 84-93: We appreciate reviewers’ comments! As reviewers’ suggestions, we added more information based on mentioned article (Dasgupta et al., 2008) in the Introduction part.

Line 58-67: Please cite references underlining developmental plasticity.

Response: Line 81-84: We appreciate reviewers’ comments! We edited the sentence and cited a reference for developmental plasticity, [13].

Section 2.1: The author may highlight the role of root cap cells on salinity tolerance with recent literature. Please reorient the section systematically following ROS accumulation, ROS scavengers and Antioxidative mechanisms, followed by tissue-specific and non-specific gene expression.: The prospects of the study/literature review may be elaborated.

Response: We appreciate reviewers' comments and suggestions to improve the manuscript. The reviewers recommended a re-organization of section 2.1 and below section 2.2, which includes systematically ROS and antioxidant mechanisms. Although we have mentioned the roles of ROS and antioxidant enzymes in the manuscript, because our focus has been mainly on the function of the root/shoot apical meristems, the current organization in this manuscript may be unfamiliar from reviewers’ viewpoint.

Our aim in this manuscript was to present a fresh perspective on salt stress, different from existing review papers. Specifically, we wanted to provide a review of various stress responses with a focus on apical meristems in terms of developmental plasticity. We believe that plant growth and development are interlinked with ever-changing environmental stresses, including salinity, in meristem-based developmental processes. Therefore, there may be differences from the conventional organization in the manuscript due to these reasons. We really ask for the generous understanding for these considerations.

Although we did not re-organize the composition, we added more information about antioxidant enzymes regulating ROS in the Introduction part (Line 86-91) and the Section2.1 part (Line 436-441).

Line 87, 106: Many publications focus on salt stress tolerance in other crops apart from Arabidopsis, please refer.

Response: We appreciate reviewers’ comments! We added and cited three more references showing the correlation between salt-tolerant growth and antioxidant enzymes in sweet potato, pea, and wheat (Dasgupta et al., 2008, [14]; Hernandez et al., 2000, [15]; Sairam and Srivastava, 2002. [16]). In addition, we added two more references showing the role of MeJA related to salt-mediated growth in bitter almond (Tavallali and Karimi, 2019. [44]), and the role of proline under salt stress in switchgrass (Guan et al., 2020, [70]). In original manuscript, we already mentioned antioxidant effect in Brassica juncea [39].

Section 2.2: Please make the sub-headings on ROS accumulation, ROS scavengers and Antioxidative mechanisms, followed by tissue-specific and non-specific gene expression.

Response: We appreciate reviewers’ comments! We have responded these comments and suggestions above.

Figure: 1A, please revise ‘palstidial GAPDH’ as ‘Plastidial GAPDH’

Response: We appreciate reviewers’ comments! We changed typo to ‘Plastidial GAPDH’ in Figure 1A. 

The authors may include another figure describing Redox signalling-Antioxidative enzyme (Enzymatic and non-enzymatic)-Metabolic action-Gene expression/regulation.

Response: We appreciate reviewers’ comments! We have responded these comments and suggestions above.

Please illustrate tissue-specific and tissue-non-specific gene regulation mechanisms of apical meristem/root upon salt stress.

Response: We appreciate reviewers’ comments! We have responded these comments and suggestions above.

The author may describe the osmoprotectants like proline upon salt stress.

Response: Line 1005-1176: We appreciate reviewers’ comments and suggestions! We added more information about the role of proline involved in stress response and development.

The role of elicitors (SA, JA, etc.) on developmental plasticity in response to salt stress may be included.: Response: Line 436-444: We appreciate reviewers’ comments! We added more information about methyl jasmonate (MeJA), which alleviated the salt-mediated inhibition of growth, such as mitotic index of root apical meristem cells and biomass of shoot/root, in bitter almond (Tavallali and Karimi, 2019). Based on this addition, we changed from ‘auxin’ to ‘phytohormones such as auxin and MeJA’, and from ‘auxin’ to ‘phytohormones’.

References may be arranged as per the journal pattern. Please cross-check the references cited with the list and include more recent relevant references.

Response: We appreciate reviewers’ comments! We added new 12 references including recent publications in the revised manuscript.

Although the overall grammar score is satisfactory, the authors are requested to polish the language, grammar, and punctuation once again.

Response: We appreciate reviewers’ comments! We checked all we could.

Additional Corrections

  • Because of the addition of new 12 references, we gave new reference numbers in the revised manuscript.
  • Line 98: remove the words ‘under salt stress conditions’
  • Line 109: edited the phrase ‘relies on the change in activities in the apical meristem niche’ to ‘relies on the activity changes in the root apical meristem (RAM)’
  • Line 174-176: edited sentence
  • Line 177: add the words ‘wild type’
  • Line 399-401: changed the word ‘level’ to ‘maxima’ and removed the repetitive words ‘through the disappearance of auxin maxima’
  • Line 407: remove ‘In addition,’
  • Line 696: changed the word ‘via’ to ‘between’
  • Line 976-980: we edited sentences. We added words ‘Because’ and ‘these’ and changed the word ‘acts’ to ‘is that they serve’
  • Line 981: Add the word ‘Meanwhile’
  • Line 1178: add the word ‘caused’
  • Line 1181: add the words ‘central zone’
  • Line 1182: add the word ‘was’
  • Line 1187-1190: edited sentence
  • Line 1206-1207: edited sentence
  • Line 1337-1338: edited sentences

Reviewer 2 Report

As stresses usually inhibit plant growth and development lowering productivity it is important to understand the molecular mechanisms through which apical meristems inhibit their activity and defense against harmful factors. The authors therefore take up a significant topic and give a lot of current data. However, the sentences are many times too long and too rich in data (reading the paper was like reading "In search of lost time") even if they are correct. Many phrases are repeated too often (e.g. intersting/ly, recent/ly) even in the title. The text is tiresomely dense.

In the 2.1 paragraph the content corresponds to the title, but the 2.2 paragraph is rather about stress- (not salt-) triggered stem meristem response - the authors are avare of this avoiding the word salt in the caption of Fig. 1B

I would suggest rewording the text.

I have noted some comments in the attached file.  

Author Response

Dear Editor and Reviewers,

Most of all, we really appreciate the thoughtful reviewers’ comments and suggestions. We have considered them carefully and followed those suggestions to address corresponding concerns and improve our manuscript. We submit two versions of revised manuscripts (marked and clean version) and one revised Figure 1. Detailed our responses to reviewers’ comments and suggestions are mentioned to each comment.

Response to Reviewer 2 Comments

ijms-2305510-peer-review-v2

As stresses usually inhibit plant growth and development lowering productivity it is important to understand the molecular mechanisms through which apical meristems inhibit their activity and defense against harmful factors. The authors therefore take up a significant topic and give a lot of current data. However, the sentences are many times too long and too rich in data (reading the paper was like reading "In search of lost time") even if they are correct. Many phrases are repeated too often (e.g. intersting/ly, recent/ly) even in the title. The text is tiresomely dense.

In the 2.1 paragraph the content corresponds to the title, but the 2.2 paragraph is rather about stress- (not salt-) triggered stem meristem response - the authors are avare of this avoiding the word salt in the caption of Fig. 1B. I would suggest rewording the text. I have noted some comments in the attached file.  

Response: We appreciate reviewers’ comments and suggestions to improve the manuscript! Because reviewers noted comments in the pdf file, we will list our response to the reviewers’ comments and suggestions in order.

  1. Title: Line 2-3: According to reviewers’ suggestions, we remove the repetitive words such as ‘in response to salt stress conditions’. Our revised title is ‘Salinity-triggered responses in plant apical meristems for developmental plasticity’.

  1. Abstracts: Line 10-15: According to reviewers’ comments and suggestions, we edited the poor English part in Abstracts through professional English service. In addition, the last sentence has been removed because there is already a summary of the manuscript in the front part.

  1. Introduction:
  • Line 21: According to reviewers’ comments and suggestions, we edited the poor English part in the first sentence of Introduction through professional English service.
  • Line 72: According to reviewers’ comments and suggestions, we changed a typo ‘SCABP’ to ‘SCABP8’.

  1. Section 2.1:
  • Line 102: According to reviewers’ comments and suggestions, we deleted the word ‘almost’.
  • Line 144: According to reviewers’ comments and suggestions, we deleted the words ‘other than roots’.
  • Line 170-171: To specify ‘In this study’ from previous mentioned two studies, we re-wrote from ‘In this study, authors’ to ‘Iglesias and colleagues’. In addition, we added the correct reference number, [32].
  • Line 405-407: According to reviewers’ comments and suggestions, we edited the poor English part in the Section 2.1 through professional English service.
  • Line 409: According to reviewers’ comments, we changed the word ‘treated’ to ‘applied’.
  • Line 425-430: According to reviewers’ comments and suggestions, we divided one sentence into three sentences with editing.

  1. Section 2.2:
  • Line 445: According to reviewers’ concerns, we edited the sub-title of the Section 2.2 to ‘Plastic growth and development in shoot meristems triggered by abiotic stress including salt’.
  • Line 710-716: According to reviewers’ comments and suggestions, we divided too long and too complicated sentence with edition.
  • Line 712: According to reviewers’ comments, we edited the phrase ‘is recognized through the mainly’ to ‘is recognized mainly through the’.
  • Line 719: According to reviewers’ comments, we changed the words ‘showed phosphorylated’ to ‘caused’.
  • According to reviewers’ comments, we removed some unnecessary words such as ‘recent/recently’ and ‘interesting/ly/intriguingly’. The current number of ‘recent/ly’ is 10 (before 20) and the number of ‘interesting/ly/intriguingly’ is 7 (before 16) in the revised manuscript.
  • Line 733: According to reviewers’ comments, we changed the word ‘represented’ to ‘shown’.
  • Line 735: According to reviewers’ comments, we changed the phrase ‘the cell division rate of clv3-2 under salt stress conditions’ to ‘it’.
  • Line 1330: According to reviewers’ comments, we changed the word ‘salt’ to ‘this’.
  • Line 1342: According to reviewers’ comments, we changed the word ‘same’ to ‘the same’.

Additional Corrections

  • Because of the addition of new 12 references, we gave new reference numbers in the revised manuscript.
  • Line 98: remove the words ‘under salt stress conditions’
  • Line 109: edited the phrase ‘relies on the change in activities in the apical meristem niche’ to ‘relies on the activity changes in the root apical meristem (RAM)’
  • Line 174-176: edited sentence
  • Line 177: add the words ‘wild type’
  • Line 399-401: changed the word ‘level’ to ‘maxima’ and removed the repetitive words ‘through the disappearance of auxin maxima’
  • Line 407: remove ‘In addition,’
  • Line 696: changed the word ‘via’ to ‘between’
  • Line 976-980: we edited sentences. We added words ‘Because’ and ‘these’ and changed the word ‘acts’ to ‘is that they serve’
  • Line 981: Add the word ‘Meanwhile’
  • Line 1178: add the word ‘caused’
  • Line 1181: add the words ‘central zone’
  • Line 1182: add the word ‘was’
  • Line 1187-1190: edited sentence
  • Line 1206-1207: edited sentence
  • Line 1337-1338: edited sentences
